# Blockade of p38 MAPK overcomes AML stem cell line KG1a resistance to 5-Fluorouridine and the impact on miRNA profiling

**Sabine Matou-Nasri**[1]*, **Maria Najdi**[1,2], **Nouran Abu AlSaud**[3], **Yazeid Alhaidan**[4], **Hamad Al-Eidi**[1], **Ghada Alatar**[1], **Deemah AlWadaani**[4], **Thadeo Trivilegio**[5], **Arwa AlSubait**[5], **Abeer AlTuwaijri**[4], **Manal Abudawood**[6], **Bader Almuzzaini**[4]*

1 Cell and Gene Therapy Group, Medical Genomics Research Department, King Abdullah International Medical Research Center, King Saud bin Abdulaziz University for Health Sciences, Ministry of National Guard-Health Affairs, Riyadh, Saudi Arabia, 2 Postgraduate program, King Saud University, Riyadh, Saudi Arabia, 3 Department of Cellular Therapy and Cancer Research, King Abdullah International Medical Research Center, King Saud bin Abdulaziz University for Health Sciences, Ministry of National Guard-Health Affairs, Riyadh, Saudi Arabia, 4 Medical Genomics Research Department, King Abdullah International Medical Research Center, King Saud bin Abdulaziz University for Health Sciences, Ministry of National Guard-Health Affairs, Riyadh, Saudi Arabia, 5 Medical Research Core Facility and Platforms, King Abdullah International Medical Research Center, King Saud bin Abdulaziz University for Health Sciences, Ministry of National Guard-Health Affairs, Riyadh, Saudi Arabia, 6 Department of Clinical Laboratory Sciences, Chair of Medical and Molecular Genetics Research, College of Applied Medical Sciences, King Saud University, Riyadh, Saudi Arabia

* matouepnasrisa@ngha.med.sa (SMN); MuzzainiB@ngha.med.sa (BA)

**Data Availability Statement:** All relevant data are within the article and its Supporting Information files.

## Abstract

Most of the AML patients in remission develop multidrug resistance after the first-line therapy and relapse. AML stem cells have gained attention for their chemoresistance potentials. Chemoresistance is a multifactorial process resulting from altered survival signaling pathways and apoptosis regulators such as MAPK, NF-κB activation and ROS production. We targeted the survival pathway p38 MAPK, NF-κB and ROS generation in human chemoresistant AML stem cell line KG1a, susceptible to enhance cell sensitivity to the chemotherapy drug 5-Fluorouridine, compared to the chemosensitive AML cell line HL60. After confirming the phenotypic characterization of KG1a and HL60 cells using flow cytometry and transcriptomic array analyses, cell treatment with the NF-κB inhibitor IKKVII resulted in a complete induction of apoptosis, and a few p38 MAPK inhibitor SB202190-treated cells underwent apoptosis. No change in the apoptosis status was observed in the ROS scavenger N-acetylcysteine-treated cells. The p38 MAPK pathway blockade enhanced the KG1a cell sensitivity to 5-Fluorouridine, which was associated with the upregulation of microribonucleic acid-(miR-)328-3p, as determined by the microarray-based miRNA transcriptomic analysis. The downregulation of the miR-210-5p in SB202190-treated KG1a cells exposed to FUrd was monitored using RT-qPCR. The miR-328-3p is known for the enhancement of cancer cell chemosensitivity and apoptosis induction, and the downregulation of miR-210-5p is found in AML patients in complete remission. In conclusion, we highlighted the key role of the p38 MAPK survival pathway in the chemoresistance capacity of the AML stem cells and potentially involved miRNAs, which may pave the way for the

**Funding:** This project was fully funded by King Abdullah International Medical Research Centre (KAIMRC) under grant number RC13/250. The funders had no role in study design, data collection and analysis, decision to publish, or preparation of the manuscript.

**Competing interests:** The authors have declared that no competing interests exist.

**Abbreviations:** ABC, ATP-binding cassette; AML, Acute myeloid leukemia; APC, allophycocyacin; BSA, bovine serum albumin; CD, cluster of differentiation; cRNA, complementary RNA; DMSO, dimethyl sulfoxide; Dox, doxorubicin; FACS, fluorescence-activated cell sorter; FCS, fetal calf serum; FITC, fluorescein isothiocyanate; FUrd, 5-fluorouridine; GAPDH, glyceraldehyde 3-phosphate dehydrogenase; IKK, inhibitor of NF-κB kinase; MDR, multidrug resistance; miRNA, microribonucleic acid; MRD, minimal residual disease; NAC, N-acetyl-L-cysteine; NF-κB, nuclear factor kappa-light-chain-enhancer of activated B cells; PBS, phosphate-buffered saline; PE, phycoerythrin; RPMI, Roswell Park Memorial Institute Medium; ROS, reactive oxygen species; RT-qPCR, reverse transcription-quantitative polymerase chain reaction; TAC, Transcriptomic Analysis Console; TBS, Tris-buffered saline.

development of a new therapeutic strategy targeting survival signaling proteins and reduce the rate of AML relapse.

## Introduction

Leukemia is the tenth leading cause of cancer-related mortality globally, with more than 474,000 new cases in 2020 and it is the fourth most prevalent cancer in the Asian and Pacific Islander populations [1, 2]. Acute myeloid leukemia (AML) is the most frequent type of leukemia affecting mainly adults, and the incidence increases with age [3]. AML also known as acute myelogenous leukemia or acute non-lymphocytic leukemia, is cancer of the myeloid line of the hematopoietic progenitors. There are eight main subtypes of AML, called $M_0$ to $M_7$, depending on the cancerous cell type and the stage of the maturation process (i.e. expression of embryonic antigens) [4, 5]. Lately, AML has also been classified based on the genomic features, chromosomal abnormalities, somatic acquired mutations, gene expression, methylation status and microribonucleic acid (miRNA) profiles [6, 7]. Patients diagnosed with AML are currently treated with the combination of chemotherapy (e.g. doxorubicin, vincristine, mitoxantrone, and methotrexate) with immunotherapy (e.g. anti-CD33 antibody) [8, 9]. However, for the treatment of severe and aggressive AML cases, this first-line regimen cannot prevent relapse even after bone marrow or allogeneic hematopoietic stem cell transplantation [10, 11]. Additionally, 70% to 80% of the AML patients achieve a complete remission; however, more than 50% develop multidrug resistance (MDR) and relapse [12, 13]. Several case studies reported that patients with relapsed AML had a unique MDR gene signature [14, 15]. Through MDR gene profiling, miRNA expression profiling and DNA methylation status have been established, most clinical trials evaluating the strategy of inhibiting MDR have been unsuccessful [7, 16, 17]. There is a growing interest in the development of therapeutic strategies for the enhancement of the chemotherapy sensitivity (chemosensitivity) of the AML cells, which would overcome the adverse risks associated with MDR and AML relapse.

The application of advances in genomic and proteomic technologies has provided molecular insights in the characterization of AML for suitable treatment. This has led to the validation of novel biomarkers for classification, risk stratification and tailored drugs suitable for AML patients [18–20]. It was also demonstrated that AML is arranged as a loose hierarchy in which a small population of self-renewing leukemic stem cells was more chemoresistant to current therapeutics than rapidly-proliferating leukemic progenitors [13, 21]. Further studies reported that MDR involves 380 genes related to metabolism, oxidative metabolism, signal transduction, DNA repair, stress response, oxidative stress, tumor suppressor activity, oncogenic transformation, apoptosis and drug efflux transporters [14, 22, 23]. The main function of the drug efflux transporter such as P-glycoprotein is to hydrolyze adenosine triphosphate to actively pump drugs out of the cell through the ATP-binding cassette (ABC) transporters [24]. Despite the varying mechanisms of action between chemotherapy and immunotherapy, the cells developing MDR become cross-resistant to immunotherapy [25, 26]. The chemoresistance process is multifactorial and involve mechanisms of drug resistance-related enzymes, genetic and miRNA alterations, resistance to apoptotic mechanisms, and aberrant activation of drug resistance-related signal pathways, including reactive oxygen species (ROS) generation, the nuclear factor-kappa B (NF-κB) pathway, and the p38 mitogen-activated protein kinase (MAPK) survival pathway [17, 27].

Defined as AML subtypes presenting different molecular characteristics, the stem cell line KG1a and promyelocytic leukemia cell line HL60 were used to evaluate their chemoresistance

potential to conventional chemotherapeutic drugs (i.e. doxorubicin, 5-fluorouridine). The importance of the p38 MAPK survival pathway, NF-κB pathway and ROS production in chemoresistance capacity was investigated using specific pharmacological inhibitors. The chemoresistance potential was evaluated by assessing cell viability, determining the apoptotic status and caspase activity, and establishing transcriptomic analysis profiling including miRNA expression profiling associated with drug resistance.

## Materials and methods

### Reagents

The culture media and reagents for cell culture were procured from Gibco® (Thermo Fisher Scientific Inc., Waltham, MA). Doxorubicin (Dox, #sc-504652), SB202190 (#sc-222294), and N-acetyl-L-cysteine (NAC, #sc-202232) were purchased from Santa Cruz Biotechnology, Inc. (Dallas, TX). Dimethyl sulfoxide (DMSO, #D2438), 5-Fluorouridine (FUrd, #F5130), inhibitor of NF-κB kinase (IKK) VII (IKKVII, #401486), and all other reagents unless otherwise mentioned were provided by Sigma-Aldrich Corp. (St. Louis, MO, USA).

### Cell culture and treatment

The human AML stem cell line KG1a (#CCL-246.1) and Caucasian promyelocytic leukemia cell line HL60 (#CCL-240) were provided by the American Type Culture Collection (Manassas, VA, USA). The cells were maintained in RPMI-1640 medium supplemented with 10% heat-inactivated fetal calf serum (FCS) and antibiotics (100 IU/mL penicillin and 100 μg/mL streptomycin). The cells were cultured at 37˚C in a 5% $CO_2$-incubator with saturated humidity. Reaching 80–90% confluence, the cells were split in a ratio of 1: 3 for each passage and the cells were then used for downstream applications between passage 4 and 9.

Both KG1a and HL60 cells were treated with various concentrations (0.001–10 μM) of conventional chemotherapeutic drugs Dox and FUrd for different incubation time periods (24, 48, and 72 h). NAC, IKKVII and SB202190 pharmacological inhibitors blocking ROS generation, NF-κB and p38 MAPK pathways, respectively, were added at the optimized concentration to the cells 2 h before the cell treatment with the chemotherapeutic drug. All chemotherapeutic drugs and pharmacological inhibitors were reconstituted in DMSO. For all the downstream applications, the cells were treated with 0.1% DMSO, the concentration corresponded to the highest volume of the pharmacological inhibitor and chemotherapeutic drug tested, which showed no toxicity or change in the cell response.

### Fluorescence-activated cell sorter (FACS) analysis

To verify the phenotype of the chemoresistance potential of KG1a AML stem cells described to be CD34+/CD38- [28], the cells were washed with phosphate-buffered saline (PBS) supplemented with 2% FCS and centrifuged at 300×$g$ for 10 min. After centrifugation, $10^5$ cells were re-suspended in 20 μL of PBS-2% FCS then 0.1 μg of mouse anti-CD34 antibody-fluorescein isothiocyanate (FITC) (#130-081-001), 0.1 μg of mouse anti-CD38 antibody-allophycocyanin (APC) (#130-092-261), IgG2a-FITC (#130-091-837) or IgG2a-APC (#130-091-836) from MACS Miltenyi Biotec (Bergisch Gladbach, Germany) were added and the mixture was kept on ice for 30 min. The excess antibodies were removed by washing the cells twice with PBS-2% FCS. The cells were pelleted then re-suspended in PBS and 10,000 cells were analyzed on a Becton Dickinson (BD) FACScanto II flow cytometer using Diva software (BD Biosciences, San Jose, CA).

The apoptosis status was determined using a BD Annexin V apoptosis detection kit (BD Biosciences) according to the manufacturer's instructions. Briefly, $10^5$ cells were pelleted and

re-suspended in 100 μL of 1× binding buffer. Five microliters of Annexin V conjugated to FITC were added to each sample and left at room temperature for 15 min in the dark. The cells were first washed with 2 mL of 1× binding buffer, pelleted and then re-suspended in 200 μL of 1× binding buffer. Five microliters of DNA dye propidium iodide (PI, viability staining solution) conjugated to phycoerythrin (PE) were added to each sample and 10,000 cells were immediately analyzed using BD Biosciences FACSDiva™ software on FACSCanto™ II flow cytometer. Early and late apoptotic cells were characterized by the binding of Annexin V to phosphatidylserine (considered as Annexin V-positive cells) without and with detection of DNA (considered as PI-negative cells and PI-positive cells, respectively) and presenting the phenotypes Annexin $V^+/PI^-$ (early apoptosis) and Annexin $V^+/PI^+$ (late apoptosis). Viable cells were characterized by Annexin $V^-/PI^-$ while necrotic cells were characterized by Annexin $V^-/PI^+$.

## RNA extraction and Transcriptomic array analysis

Total RNA was extracted using a RNeasy Plus Mini kit (Qiagen, Hilden, Germany) according to the manufacturer's recommendations and subjected to the Clariom™ D Human Transcriptome, Affymetrix Genechip® miRNA 4.0 array (Affymetrix, Santa Clara, CA, USA), and to the miRNA reverse transcription-quantitative polymerase chain reaction (RT-qPCR).

## Transcriptomic array analysis

The transcription profiling was established using the Clariom™ D Human Transcriptome (Affymetrix) following manufacturer's instructions. In brief, 100 ng of total RNA were extracted from a triplicate of each experimental group of KG1a and HL60 cell lines to prepare biotinylated cRNA fragments, which were hybridized to the Clariom™ D Human Transcriptomic array. Hybridization and washing were performed using GeneChip® hybridization oven 645, GeneChip® fluidics station 450 and GeneChip® scanner 3000 7G (Thermo Fisher Scientific Inc.). The difference in gene expression between the two samples were determined using the Transcriptomic Analysis Console (TAC) software version 4 (Thermo Fisher Scientific Inc.).

The miRNA microarray analysis was performed using the Affymetrix Genechip® miRNA 4.0 array after total RNA extraction from the KG1a cells that were non-treated or pre-treated with 10 μM SB202190 followed by the presence or absence of 1–10 μM 5-FUrd for 48 h of incubation. Briefly, the whole RNA extract was labeled using the Affymetrix FlashTag™ Biotin HSR RNA Labeling Kit. After being quantified using the NanoDrop® ND-1000 spectrophotometer (Marshall Scientific, Hampton, NH, USA), labeled RNA was hybridized to the Affymetrix Genechip® miRNA 4.0 array according to the manufacturer's protocol and scanned using the GeneChip® Scanner 3000 7G. The signal value assessment was determined using the Affymetrix Genechip® Command Console software. U6 small nuclear RNA was used as a reference gene.

## Cell viability assay

The cell viability assay was performed after exposing the cells to the conventional chemotherapeutic drugs Dox and FUrd at different doses (0.001–10 μM) for 72 h incubation using the Promega CellTiter-Glo® assay kit (Madison, WI, USA), according to manufacturer's instructions. Briefly, the cell viability assessment was based on the quantification of the ATP present, a molecular indicator of metabolically active cells. The CellTiter-Glo® assay generated a "glow-type" luminescent signal produced by luciferase, proportional to the percentage of living cells. The luminescent signal was measured using EnVision microplate reader (Perkin Elmer, Waltham, MA, USA).

## Protein extraction and Western blot analysis

The cell lysates from the non-treated and treated cells were prepared by adding 80 μl of NP40 lysis buffer (Invitrogen, Thermo Fisher Scientific Inc.) as previously described in [29]. The extracted proteins were estimated as per the Qubit® Protein assay kit protocol (Invitrogen) and the sample protein concentrations were read by the Qubit® fluorometer (Invitrogen). From the cell lysate preparation to the protein separation by 12% sodium dodecyl sulfate-poly-acrylamide gel electrophoresis, western blot technology was performed as described in [30]. The polyvinylidine difluoride membranes were stained with the following primary antibodies (Abcam, Cambridge, UK) diluted in blocking buffer [1% bovine serum albumin (BSA) in Tris-buffered saline (TBS)–Tween 20]: rabbit and mouse monoclonal antibodies against the phos-pho- (Y182) (#ab47363) and total (#ab31828) forms of p38 MAPK, and against α-Tubulin (# ab18251), at 4˚C overnight with continuous shaking. The primary antibodies were detected with infrared fluorescent IRDye® 680RD goat anti-rabbit and IRDye® 800RD goat anti-mouse secondary antibodies (LI-COR Biosciences, Lincoln, NE, USA) diluted in TBS–Tween containing 3% BSA (1:1,000) for 1 h at room temperature with continuous mixing. After five washes in TBS–Tween, the proteins were visualized using the LI-COR Odyssey CLx Scanner (LI-COR Biosciences) and analyzed using ImageJ software v1.46r (https://imagej.nih.gov/ij/download.html).

## Confocal laser scanning microscopy

The Caspase-3/-7 activity and the mitochondrial permeability transition pore activity were assessed using the Invitrogen Image-iT® LIVE Red Caspase-3 and Caspase-7 Detection Kit and the Image-iT® LIVE Mitochondrial Transition Pore Assay Kit according to the manufac-turer's instructions. The caspase activity was indicated by the red fluorescence and an increased mitochondrial outer membrane permeabilization was indicated by the quenched green fluorescence of the calcein. The photomicrographs were taken using an LSM780 confo-cal laser scanning microscope (Carl Zeiss Microscopy GmbH, Jena, Germany).

## cDNA production and miRNA RT-qPCR

After the RNA extraction, the cDNA templates were synthesized using a TaqMan Advanced miRNA cDNA synthesis kit (#A28007) according to the manufacturer's protocol (Thermo Fisher Scientific Inc.). The expression level of miR-210-5p, miR-26b-5p and miR-328-3p was evaluated using Thermo fisher scientific TaqMan Advanced miRNA assays following manu-facturer's instructions, including pre-designed primer sequences as follows: 5ʹ–CUGUGCGU GUGACAGCGGCUGA–3ʹ for miR-210-5p (#477970mIR), 5ʹ–UUCAAGUAAUUCAGGAUAGG U–3ʹ for miR-26b-5p (#478418miR), and 5ʹ–CUGGCCCUCUCUGCCCUUCCGU–3ʹ for miR-328-3p (#4788028miR). The miR transcript expression levels were normalized to endogenous control gene miR-26b-5p (5'-UUAUCAGAAUCUCCAGGGGUAC-3', #478056miR). A real-time PCR using the parameters of 95˚C for 20 sec, followed by 40 cycles of 95˚C for 1 sec and 60˚C for 20 sec was performed using Applied Biosystem™ QuantStudio 6 Flex system on the 7500 Real-time PCR system (Thermo Fisher Scientific Inc.).

## Statistical analysis

The data are expressed as mean ± standard deviation (SD) for the experiments performed in triplicate and each experiment was independently repeated three times. Statistical differences were estimated using One-way ANOVA test with GraphPad Prism software. Values of $p < 0.05$ were considered significant. Expression array feature intensity (CEL) files were

analyzed using the Affymetrix Transcriptome Analysis Console (TAC) software version 4.0. Differential gene expression was determined by minimum 2.0-fold change, median false discovery rate (FDR) < 0.05 and $p$-value < 0.05 was considered significant.

## Results

### Assessment of the resistance potential of the AML cell lines KG1a and HL60 exposed to the chemotherapeutic drugs doxorubicin and 5-fluorouridine

The chemoresistance potential of the AML stem cell line KG1a is associated with the phenotype CD34+/CD38- [28]. After immuno-staining of the KG1a cells with anti-CD34-FITC and anti-CD38-APC monoclonal antibodies, the cells were analyzed with the flow cytometer using FACS Diva software. Almost all the cells were positive for the stem cell marker CD34 and negative for the lymphocyte differentiation marker CD38, compared with the unstained cells and cells immuno-stained with the isotype controls (Fig 1A). To highlight the main gene expression profiling characterizing the chemoresistance potential of the AML stem cell line KG1a, differentially expressed genes in the KG1a cells in comparison with the promyelocytic leukemia HL60 cells were analyzed using the Affymetrix Clariom™ D Human Transcriptomic array after total RNA extraction from the untreated cultured KG1a cells and HL60 cells. A classification of the 100 genes most differentially expressed (top upregulated and downregulated, S1 Table) by comparing the KG1a (resistant) against the HL60 (sensitive) cell lines was arranged

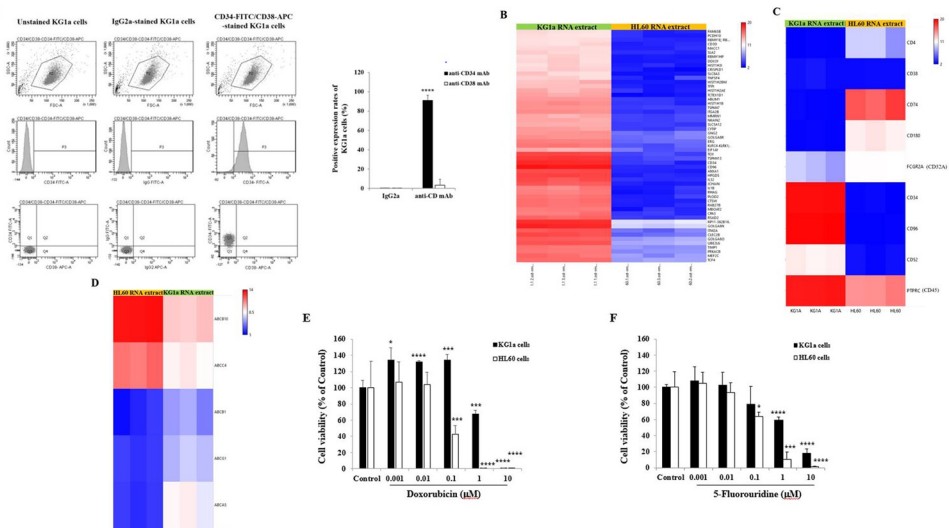

**Fig 1. Characterization of the chemoresistance potential of leukemic stem-like AML cell line KG1a.** (A) Representative cell scatter plots and histogram analysis of leukemic stem-like AML cell line KG1a after immuno-cyto-staining with FITC-labeled anti-CD34 antibody, with APC-labeled anti-CD38 antibody or with respective IgG2a-FITC and IgG2a-APC isotype controls confirming the phenotypic characteristic of the chemoresistance potential of stem cells defined as CD34+/CD38-. Bar graph showing the positive expression rates of KG1a cells based on CD34 and CD38 cell surface detection. (B) Transcriptomic profiling and analysis of differentially expressed genes in AML KG1a stem cells and HL60 cells. Using the Clariom™ D Human Transcriptomic array, selection of the top 100 differentially expressed genes. (C) KG1a and HL60 cell characterization based on selected transcriptomic profiling using the ClariomD™ array. (D) Highlight on the top differentially expressed drug resistance-related genes in HL60 and KG1a cells. (E, F) Bar graphs showing the percentage viability of the KG1a and HL60 cells treated with various concentrations (0.001–10 μM) of either Doxorubicin (D) or 5-Fluorouridine (E). After 72 h incubation, the cell viability was determined using CellTiter-Glo® and expressed as percentage of the control, the non-treated cell viability, corresponding to 100%. The results are presented as mean ± SD and were repeated through three independent experiments. *$p$ < 0.05, ***$p$ < 0.001, and ****$p$ < 0.0001 vs. control.

(Fig 1B). As expected for the stem cells, the leukemic stem cell-specific markers such as cluster of differentiation (CD)34 and CD96 [31] gene expression levels were extremely high in the KG1a cells, with the expression levels extremely low in the HL60 cells (Fig 1C, S2 Table). Described as the leucocyte common antigen, the protein tyrosine phosphatase PTPRC, also known as CD45, was expressed in both the KG1a and HL60 cells but higher in the KG1a cells and the CD4, a specific marker for T cells, was weakly expressed in both the KG1a and HL60 cells (Fig 1C). Interestingly, CD74, a high-affinity glycoprotein receptor and chaperone to the MHCII molecule [32], was highly expressed in the HL60 cells and weakly expressed in the KG1a cells (Fig 1C). The most prevalent marker on most of the peripheral T and B cells, but highly expressed in KG1 cells and lowly expressed in HL60 cells [33], the transcriptomic analysis confirmed this differential CD52 expression levels in both the cell lines KG1a and HL60 (Fig 1C). The other marker CD32a, also known as FcγRIIa, whose upregulation is correlated with leukemia differentiation [34], was highly expressed in HL60 cells and weakly in the KG1a cells (Fig 1C).

Regarding the expression of the genes associated with drug resistance, a slight increase in the expression levels of the genes coding for the ATP-binding cassette (ABC) transporter proteins such as ABCB1 (P-glycoprotein), ABCG1 and ABCA5 monitored in KG1a RNA extracts was observed compared to those measured in the HL60 cells (Fig 1D, S3 Table). Of note, the gene expression levels of ABCB10 and ABCC4 were higher in the HL60 cells than in the KG1a cells (Fig 1D).

After 72 h of AML cell exposure to the conventional chemotherapeutic drugs Dox and FUrd, the viability of both KG1a and the promyelocytic leukemia HL60 cells was assessed based on ATP generation, a metabolic parameter proportional to the number of viable cells. Compared to the untreated KG1a cells (the Control), with 100% cell viability, the addition of low Dox concentrations (from 0.001 to 0.1 μM) significantly increased (+30%, $p < 0.05$) the KG1a cell viability (Fig 1E). Compared to the untreated KG1a cell viability, at 1 μM of Dox, a drastic reduction (-32.6%, $p = 0.0002$) in the KG1a cell viability was observed followed by a total cell death caused by the addition of 10 μM of Dox (Fig 1E). In terms of the HL60 cells, low concentrations (from 0.001 to 0.1 μM) of Dox did not affect the cell viability, compared to non-treated HL60 cells (Fig 1E). However, in contrast to KG1a cells, a total HL60 cell death was observed in the presence of lower Dox concentrations (from 1 μM), compared to untreated HL60 cells (Fig 1E). In contrast to Dox, the KG1a cell exposure to low FUrd concentrations (from 0.001 to 0.1 μM) did not affect KG1a cell viability, compared to the non-treated KG1a cells (Fig 1F). Compared to the healthy untreated cells, a decrease of 40% in the KG1a cell growth was observed by the addition of 1 μM of FUrd and 80% decrease in the KG1a cell viability occurred from 10 μM of FUrd (Fig 1F). In contrast to the KG1a cells, 40% and 90% decrease in the HL60 cell viability was observed by the addition of 0.1 and 1 μM of FUrd, and a total inhibition of the cell viability was obtained from 10 μM of FUrd, confirming the chemosensitive potential of HL60 cells (Fig 1F).

## Influence of the blockade of ROS production, NF-κB and p38 MAPK pathways on the induction of apoptosis in KG1a and HL60 cells

The production of ROS [35] and the activation of the NF-κB pathway [36] and the p38 MAPK pathway [37] contribute to cell survival and chemoresistance, suggesting their potential prominent role in the KG1a cell chemoresistance capacity. Both the KG1a and the HL60 cells were pre-treated with pharmacological inhibitors, such as NAC for ROS production inhibition, IKKVII and SB202190 for the NF-κB and p38 MAPK pathway inactivation, respectively. As a main cell death mechanism, the induction of apoptosis in the non-treated and treated cells was

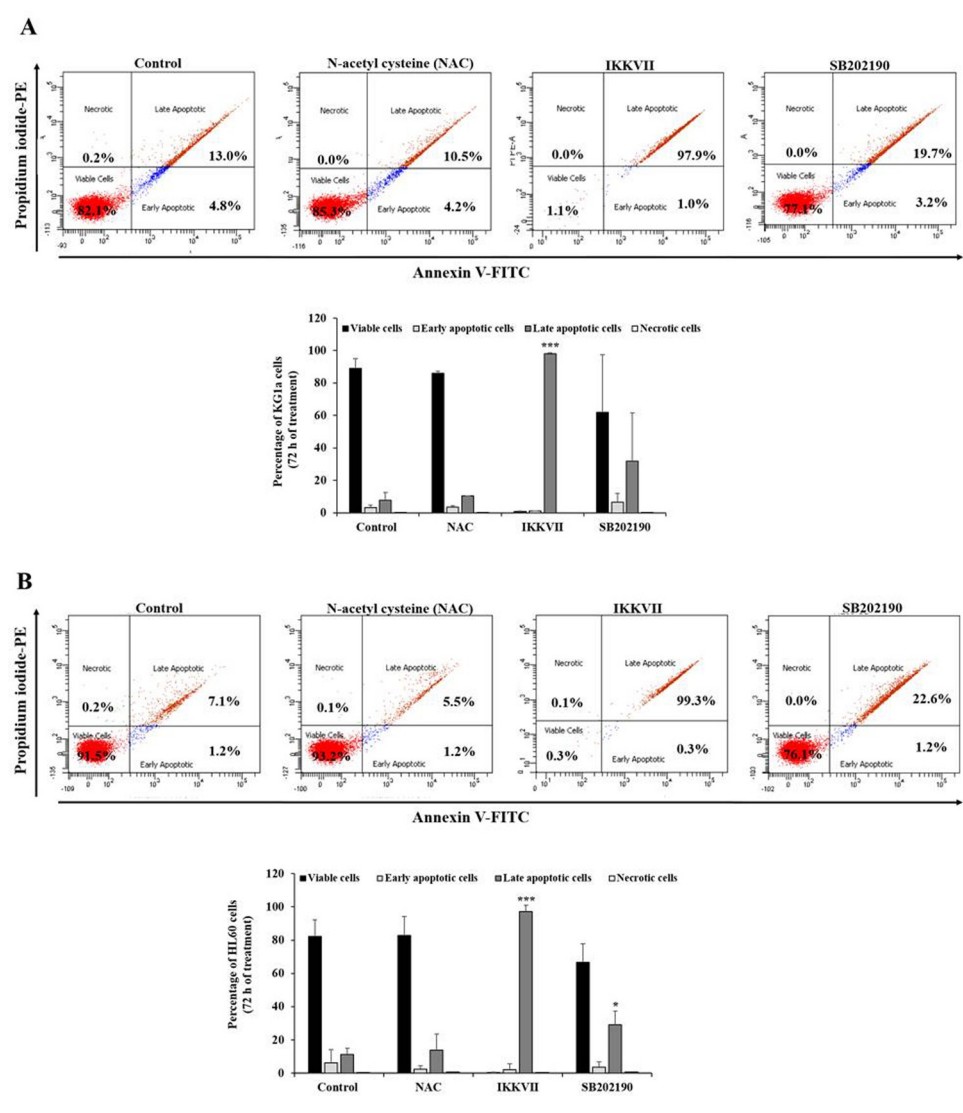

**Fig 2. KG1a and HL60 apoptotic status after cell treatment with NAC (ROS production inhibitor), IKKVII (NF-kB pathway inhibitor), and SB202190 (p38 MAPK pathway inhibitor).** The KG1a (A) and HL60 (B) cells were treated with 5 mM NAC, 20 μM IKKVII and 10 μM SB202190. After 72 h incubation, viable, (early and late) apoptotic, and necrotic status of the cells were determined using Apoptosis determination kit. Representative cell scatter plots indicating the percentage of cells determined at each status. Bar graphs showing the results presented as mean ± SD, based on three independent experiments. $^{*}p < 0.05$, $^{***}p < 0.001$, and $^{****}p < 0.0001$ *vs.* control.

determined based on the Annexin V-FITC/PI-PE double staining after 72 h incubation. The KG1a cell pre-treatment with the ROS scavenger NAC tested at a concentration of 5 mM reported to prevent ROS generation [38, 39], no significant change of the apoptotic status of the KG1a cells was observed, compared with the apoptotic status determined in the untreated KG1a cells (Fig 2A). The KG1a cell treatment with 20 μM IKKVII resulted in the induction of apoptosis, displaying an 85% increase of late apoptotic cells, and 10 μM SB202190 prompted apoptosis, revealing a 7% increase of late apoptotic cells, as compared with apoptotic status determined in the untreated KG1a cells (Fig 2A). For the HL60 cells, similar pro-apoptotic effects as in the KG1a cells were observed in the HL60 cells, which displayed a higher apoptosis percentage, including 99% and 20% of the cell population in late apoptosis, reached after the

cell treatment with IKKVII and SB202190, respectively (Fig 2B). Of the three targeted key players of apoptosis induction, only the p38 MAPK pathway inhibition had a slightly increased apoptotic status in both the KG1a and HL60 cells, which allowed a deeper investigation of its potential impact in AML cell resistance capacity once combined with the chemotherapeutic drug, FUrd.

## Potentiation of 5-fluorouridine-induced KG1a apoptosis after p38 MAPK blockade

Before investigating the impact of the p38 MAPK blockade on the FUrd-induced cytotoxicity in both the KG1a and HL60 cells based on the induction of apoptosis, the p38 MAPK phosphorylation level was assessed in the cells pre-treated with SB202190 and treated with 10 μM of FUrd. After 72 h incubation, the cell lysates were subjected to a Western blot analysis for the protein expression detection of the phospho-p38 and the total p38 MAPK. As expected, the SB202190 significantly decreased the p38 MAPK phosphorylation in both KG1a (-62.3%, $p = 0.049$) and HL60 (-65.1%, $p = 0.0053$) cells with no effect observed after the cell treatment with FUrd, compared with the basal level of p38 MAPK detected in the non-treated cells (Fig 3). The KG1a and HL60 cell treatment with FUrd following the cell pre-treatment with SB202190 resulted in a higher inhibition of the p38 MAPK phosphorylation level than caused by SB202190 alone (Fig 3).

Regarding the apoptosis status, after 24 h incubation, the FUrd significantly prompted apoptosis in the HL60 cells (~25% of late apoptotic cells, $p = 0.049$) with no effect observed after the cell treatment with SB202190 alone, compared to the non-treated cells (Fig 4A). The combination of the HL60 cell pre-treatment with SB202190, followed by the addition of FUrd, resulted in an increase in the late apoptosis percentage (~40%, $p = 0.0037$), compared to the non-treated cells (Fig 4A). However, after 72 h incubation, there was a higher percentage of

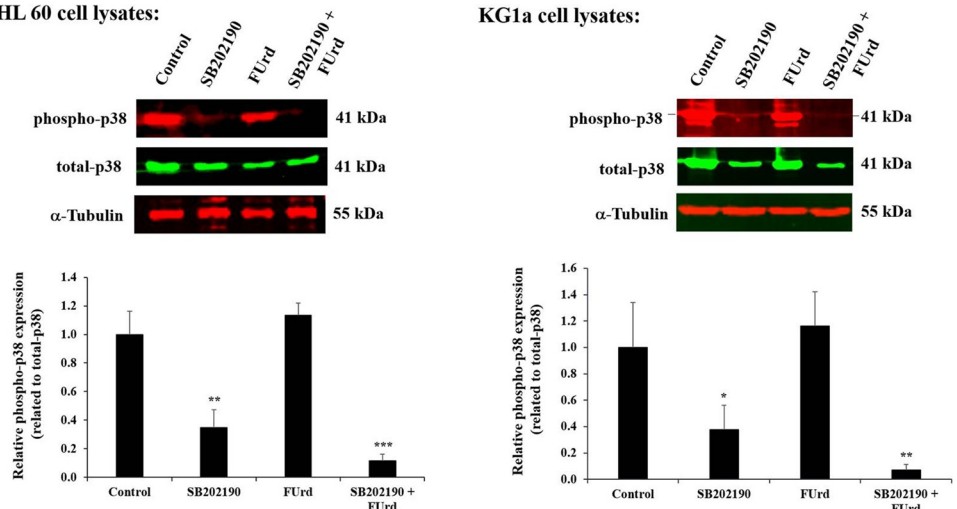

**Fig 3. Decrease of p38 MAPK phosphorylation in HL60 and KG1a cells exposed to 5-Fluorouridine (FUrd) after cell pretreatment with p38 MAPK pathway inhibitor SB202190.** The KG1a and HL60 cells were pre-treated with 10 μM SB202190 for 2 h incubation then followed by the cell treatment with 10 μM of FUrd. After 72 h incubation, the cell lysates were subjected to Western blot analysis. Representative Western blot showing the decrease of the p38 phosphorylation by SB202190, confirming the blockade of the p38 MAPK pathway. The bar graphs show the relative protein expression levels of phospho-p38 calculated as a ratio of total p38 expression (the loading control). The results are presented as mean ± SD of three independent experiments. $^{*}p < 0.05$, $^{**}p < 0.01$, and $^{***}p < 0.001$ *vs.* control. Tubulin was used as a loading control.

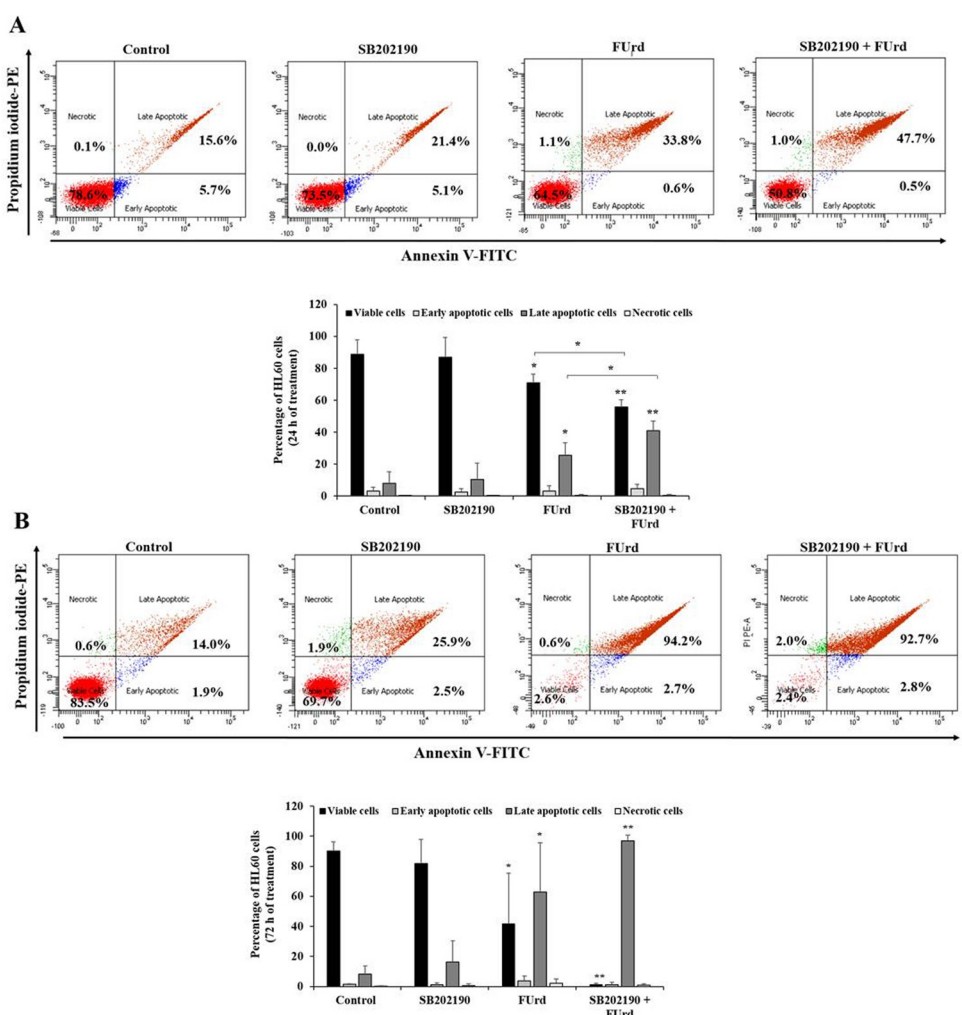

**Fig 4.** Analysis of apoptotic status in HL60 cells in response to 24-h (A) and 72-h (B) treatment with FUrd after cell pretreatment with SB202190. The HL60 cells were non-treated, treated with pharmacological inhibitors for 2 h incubation prior to cell treatment with 10 µM of FUrd. After 24 h (A) and 72 h (B) incubation, the cells underwent FITC-labeled Annexin V/PE-labeled PI double-staining for apoptosis analysis using FACS. The numbers within the scatter plots represent the percentage of viable (lower left, Annexin V$^-$/PI$^-$), early apoptotic (lower right, Annexin V$^+$/PI$^-$), late apoptotic (upper right, Annexin V$^+$/PI$^+$), and necrotic cells (upper left, Annexin V$^-$/PI$^+$). The bar graphs present the percentage viable, early apoptosis, late apoptosis and necrosis of the SB202190-pretreated HL60 cells in the presence or absence of 10 µM FUrd along with the non-treated (control) cells, based on three independent experiments. The results are presented as mean ± SD. $^*p < 0.05$, $^{**}p < 0.01$, and $^{****}p < 0.0001$ *vs.* control.

apoptotic cells in the FUrd-treated HL60 cells (~60%, $p = 0.046$) and this late apoptosis percentage was increased (~95%, $p = 0.00002$) through the addition of FUrd after the SB202190 cell treatment, compared to the non-treated cells and the SB202190-treated cells (Fig 4B). For the KG1a cells, no change of the apoptosis status was observed after 24 h of cell treatment with SB202190, FUrd or the combination, compared to the low apoptosis status in the non-treated cells (Fig 5A). However, after 72 h incubation, the FUrd-induced KG1a apoptosis prompted an increase of ~10% ($p = 0.001$) of the cell population in early apoptosis and ~35% ($p = 0.07$) in late apoptosis, compared with the non-treated KG1a cells. However, the combination of the KG1a cell pre-treatment with SB202190, followed by the addition of FUrd, resulted in an increase of the cell population in late apoptosis (~72.5%, $p = 0.00087$), a hallmark of the

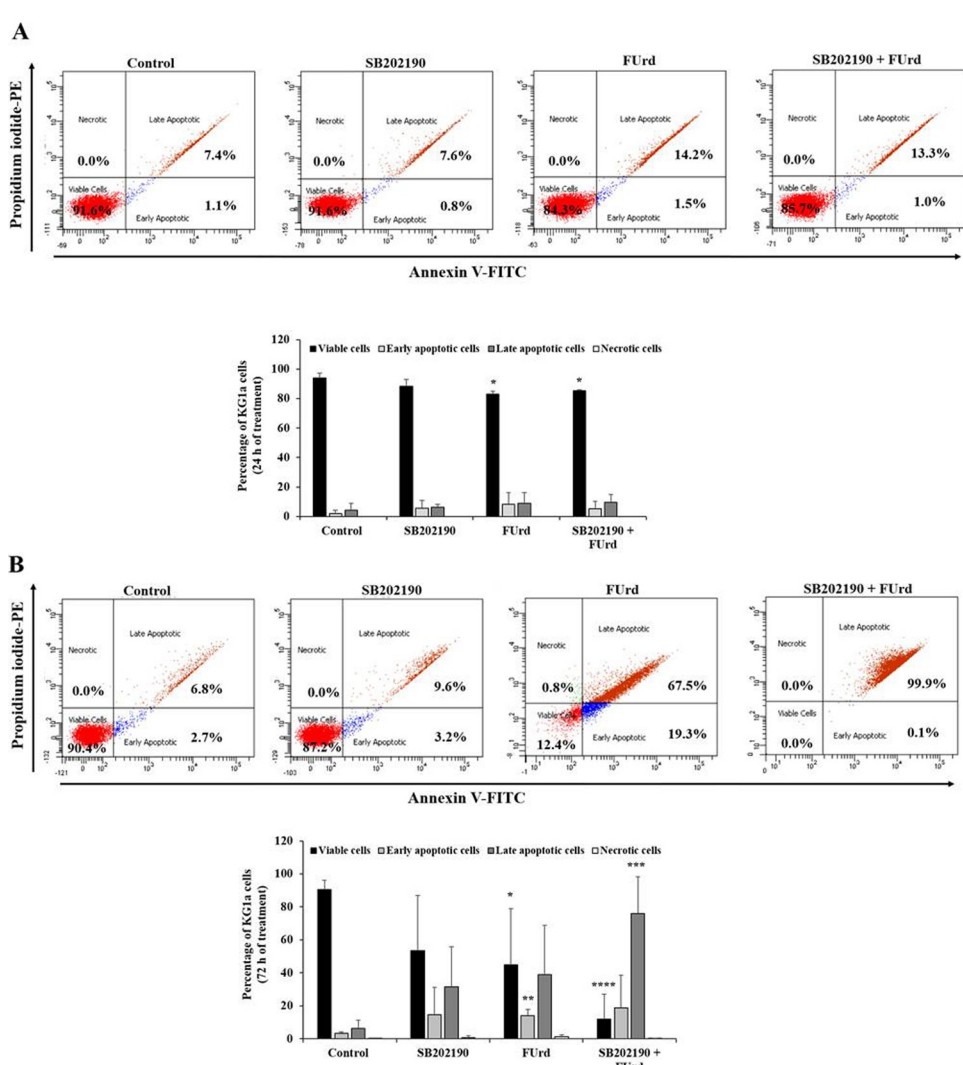

**Fig 5.** Analysis of apoptosis status in KG1a cells in response to 24-h (A) and 72-h (B) treatment with FUrd after cell pretreatment with SB202190. The KG1a cells were non-treated, treated with pharmacological inhibitors for 2 h incubation prior to cell treatment with 10 μM of FUrd. After 24 h (A) and 72 h (B) incubation, the cells underwent FITC-labeled Annexin V/PE-labeled PI double-staining for apoptosis analysis using FACS. The numbers within the scatter plots represent the percentage of viable (lower left, Annexin V⁻/PI⁻), early apoptotic (lower right, Annexin V⁺/ PI⁻), late apoptotic (upper right, Annexin V⁺/PI⁺), and necrotic cells (upper left, Annexin V⁻/PI⁺). The bar graphs present the percentage viable, early apoptosis, late apoptosis and necrosis of SB202190-pretreated KG1a cells in the presence or absence of 10 μM FUrd along with the non-treated (control) cells, based on at least three independent experiments. The results are presented as mean ± SD. $^*p < 0.05$, $^{**}p < 0.01$, $^{***}p < 0.001$, and $^{****}p < 0.0001$ *vs.* control.

enhancement of the chemosensitivity of the KG1a cells after the p38 MAPK pathway blockade (Fig 5B).

The induction of apoptosis in the HL60 cells (observed after 24 h incubation) and the KG1a cells (observed after 72 h incubation) potentiated by the combined SB202190 and FUrd treatment was associated with an enhancement of the increased activities of the effector poly-caspase-3/-7 and of the mitochondrial outer membrane permeabilization, compared with their activity prompted by the FUrd alone (Fig 6). Of note, a slight increase of the poly-caspase-3/7

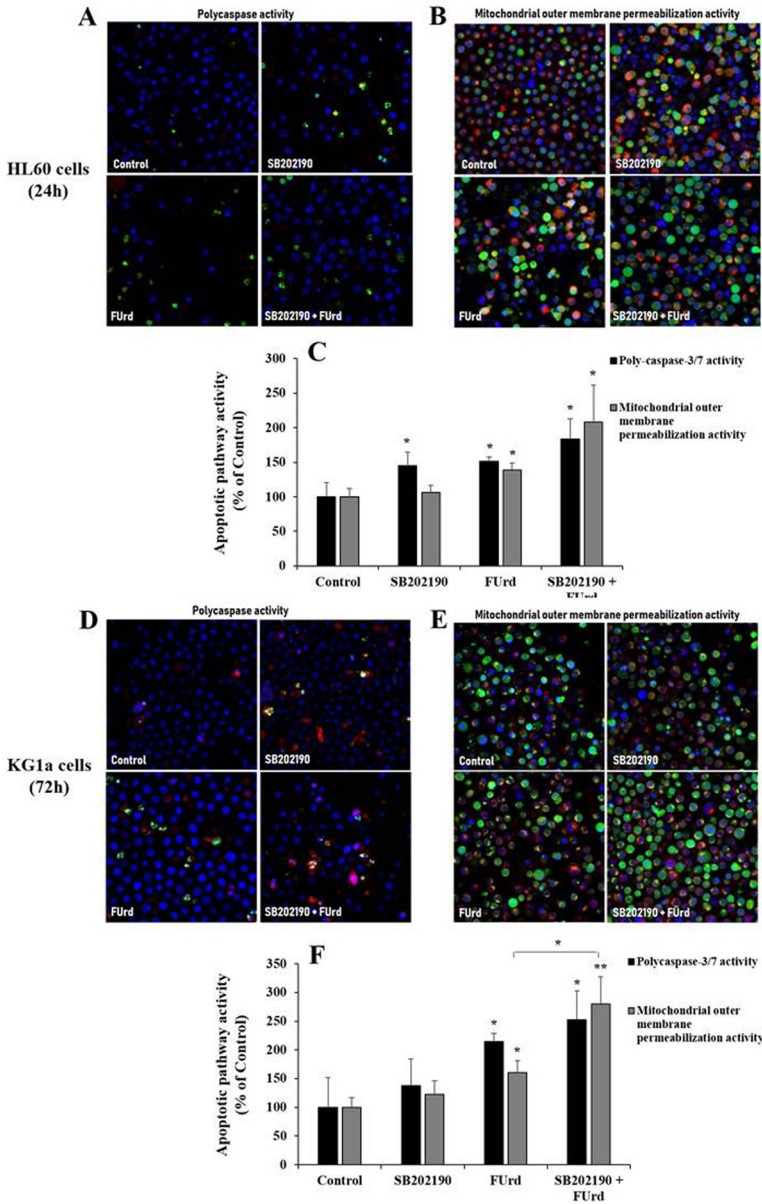

**Fig 6.** Induction of poly-caspase activity and mitochondrial outer membrane permeabilization activity in the HL60 cells (A-C) and in the KG1a (D-F) in response to FUrd after pretreatment with SB202190. Both the HL60 and KG1a cells were non-treated, treated with SB202190 for 2 h incubation prior to cell treatment with 10 μM of FUrd. After 24 h incubation for the HL60 cells (A-C) and 72 h incubation for the KG1a cells (D-F), the cells underwent staining for assessment of poly-caspase activity (A, D) and of mitochondrial outer membrane polymerization (B, E) in apoptotic cells (indicated in green). The bar graphs present the percentage of the poly-caspase activity and of the mitochondrial outer membrane polymerization determined in SB202190-pretreated HL60 (C) and KG1a (F) cells in the presence or absence of FUrd along with the non-treated (control) cells. The results are presented as mean ± SD of three independent experiments. $^{*}p < 0.05$ and $^{**}p < 0.01$ *vs.* control.

activity was noticed in the SB202190-treated HL60 cells while SB202190 did not affect the mitochondrial outer membrane permeabilization in the HL60 cells or both poly-caspase-3/7 and mitochondrial polymerization activities in the KG1a cells, as compared to the non-treated cells (Fig 6).

**Table 1. Profiling of the top differentially dysregulated drug resistance-associated miRNA in AML stem cell line KG1a exposed to SB202190 p38 MAPK inhibitor and to 1–10 μM FUrd.**

| Experimental conditions | miRNA Transcript ID | MDR-related target genes | Fold-change | *p* value |
|---|---|---|---|---|
| **SB202190 *vs.* Control** | hsa-miR-24-2-5p | ABCA1; ABCA10; ABCA12; | 2.03 | |
| | hsa-miR-30a-5p | ABCA12; ABCA9; ABCB10; ABCC10; ABCG5; | 2.27 | |
| | hsa-miR-139-5p | ABCA6; ABCB7; ABCC11; | 2.31 | |
| | hsa-miR-210-5p | ABCA4; ABCC1; | 3.69 | |
| | hsa-miR-210-3p | ABCB9; ABCC1; ABCD1; ABCD4; | 2.21 | |
| | hsa-miR-145-5p | ABCA4; ABCC1; | 2.86 | |
| | hsa-miR-335-3p | ABCA6; ABCC11; | 2.11 | |
| | hsa-miR-196b-3p | ATP6AP1; MAP2K2; | 2.17 | |
| **SB202190 + 1 μM FUrd *vs.* 1 μM FUrd** | hsa-miR-328-3p | ABCB8; ABCG2; | 3.36 | 5.89E-05 |
| | hsa-miR-139-5p | ABCA6; ABCB7; ABCC11; | 2.15 | 0.0002 |
| | hsa-miR-21-5p | ABCA1; ABCD3; | -2.3 | 0.029 |
| | hsa-miR-3613-5p | AFAP1; ANP32B; ARID1A; CDK6; COG6; | 2.59 | 0.0419 |
| **SB202190 + 10 μM FUrd *vs.* 1 μM FUrd** | hsa-miR-26b-5p | ABCA1; ABCA6; ABCB6; ABCB9; ABCC11;.. | -3.76 | 0.002 |
| | hsa-miR-210-5p | ABCA4; ABCC1; | 3.64 | 0.003 |
| | hsa-miR-7111-5p | ABCC6; | -2.05 | 0.008 |

## miRNA transcriptomic profiling in FUrd-treated "chemosensitive" KG1a cells after p38 MAPK pathway blockade

The miRNAs, regulators of the target gene expression at the post-transcriptional level, play a key role in AML chemotherapeutic resistance and their expression profiles predict cancer cell resistance potential [40, 41]. The microarray-based miRNA transcriptomic profiling was determined using the miRNA 4.0 array in the chemoresistant KG1a cells and the considered chemosensitive KG1a cells (after the p38 MAPK pathway blockade) following to their exposure to FUrd, after 48 h incubation. From Table 1, a set of miRNA transcripts was differentially expressed in the chemoresistant KG1a cells, compared to those expressed in the chemosensitive SB202190-treated KG1a cells. Considering the significant changes in the gene expression level above the standard 2.0-fold threshold, the highest upregulation of the miRNA transcript level monitored in the SB202190-treated KG1a cells corresponded to the hsa-miR210-5p (+-3.69-fold), compared to the hsa-miR210-5p basal expression level monitored in the non-treated KG1a cells (Tables 1 and S4). In response to the FUrd, another set of miRNAs transcript levels were upregulated including the hsa-miR-328-3p (+3.36-fold; 1 μM FUrd, S5 Table) and the hsa-miR-210-5p (+3.64-fold; 10 μM FUrd, S6 Table). The other miRNAs transcript levels were downregulated, such as the hsa-miR-26b-5p (-3.76-fold; 10 μM FUrd) monitored in the chemosensitive SB202190-treated KG1a cells, compared to the chemoresistant KG1a cells (Table 1). Hsa-miR-328-3p is associated with apoptosis induction and the enhancement of cell sensitivity [42], and hsa-miR-210-5p is associated with a poor prognosis when highly expressed and weakly expressed in AML patients in complete remission [43]. Hsa-miR-26b-5p is involved in AML pathogenesis. An upregulation of the hsa-miR-328-3p transcript level monitored in the chemosensitive SB202190-treated KG1a cells exposed to 10 μM FUrd was confirmed using RT-qPCR (Fig 7). Although the RT-qPCR results confirmed the upregulation of the hsa-miR-210-5p transcript level in the SB202190-treated KG1a cells, in contrast to the miRNA analysis, a significant decrease of the hsa-miR-210-5p transcript level was found in the chemosensitive SB202190-treated KG1a cells exposed to 10 μM FUrd, with no change in the hsa-miR-26b-5p transcript level, compared to the FUrd-treated chemoresistant KG1a cells (Fig 7).

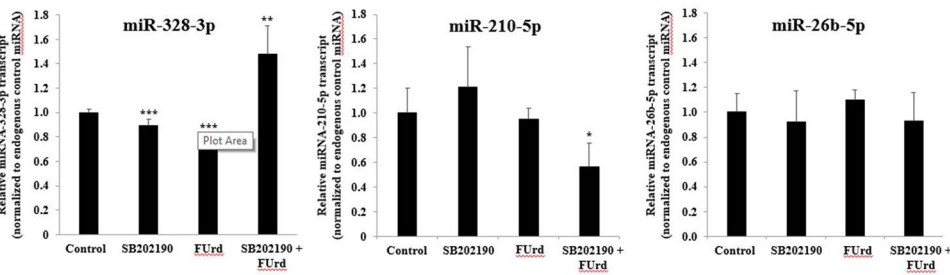

**Fig 7. miRNA transcript levels in chemoresistant and SB202190-treated chemosensitive KG1a cells exposed to FUrd using RT-qPCR.** The KG1a cells were non-treated or treated with 10 μM SB202190 for 2 h incubation prior to cell treatment with 10 μM FUrd. After 48 h incubation, the total RNA was extracted for the miRNA RT-qPCR assay. The bar graphs show the relative expression level of miR-328-3p, miR-210-5p and miR-26b-5p mRNA determined by RT-qPCR analysis in non-treated and treated KG1a cells calculated as a ratio of the expression to the endogenous control miRNA. The results are presented as mean ± SD of three independent experiments. $^*p < 0.05$, $^{**}p < 0.01$, and $^{***}p < 0.001$ *vs.* control.

## Discussion

Relapsed AML occurs in half of younger patients and in most of the elderly patients due to the development of chemoresistance capacity towards antineoplastic agents, especially from AML stem cells [11, 13]. This chemoresistance capacity is a complex process prompted by the excessive activation of the survival pathways, mitochondrial activity, and even by microRNA (miRNA) alterations [17]. There is an increasing interest in establishing a therapeutic strategy to overcome relapsed AML through the enhancement of the sensitivity of the AML stem cells to chemotherapeutic drugs. In the current study, we aimed to pinpoint prominent survival pathways activated in chemoresistant AML cells and its blockade, which may enhance their chemosensitivity and dysregulate the miRNA expression profiling. The chemoresistant human AML stem cells KG1a and the chemosensitive AML promyelocytic leukemia HL60 cells were used for the evaluation of the resistance capacity of the KG1a cells to conventional chemotherapeutic drugs (i.e. doxorubicin, Dox; 5-fluorouridine, FUrd) using specific pharmacological inhibitors such as SB202190, IKK inhibitor VII, and NAC for the blockade of the p38 MAPK survival pathway, NF-κB pathway and ROS production, respectively. The chemoresistance potential was determined by assessing the cell viability, monitoring the apoptosis status and establishing a microarray-based miRNA transcriptomic profiling analysis associated with AML patient outcomes in response to conventional chemotherapy. FACS and Clariom™ D transcriptomic array analyses confirmed the stem cell features of the AML KG1a and the promyelocytic leukemia characteristics of the HL60 cells. The treatment of both the AML cell lines with the NF-κB inhibitor IKKVII resulted in a complete induction of apoptosis with no changes after the addition of the ROS scavenger NAC, compared to the non-treated cells. However, the blockade of the p38 MAPK pathway resulted in a slight increase of the apoptotic cell percentage but concomitantly enhanced the KG1a cell sensitivity when exposed to FUrd. Using microarray-based miRNA transcriptomic and RT-qPCR analyses, an upregulation of the transcript level of miR-328b-3p, associated with cancer cell chemosensitivity, and a downregulation of the transcript level of the miR-210-5p reported in AML patients declared in complete remission, were monitored in the chemosensitive KG1a cells exposed to FUrd. We highlighted the key role of the p38 MAPK survival pathway in the chemoresistance capacity of the AML stem cells, which may pave the way for the development of a new therapeutic strategy targeting survival-related signaling proteins and reduce the AML relapse rate.

The cytotoxic effects of the Dox and FUrd on the AML cell lines KG1a and HL60 were assessed using a cell viability assay. The addition of low concentrations of either Dox or FUrd

maintained the KG1a cell in high viability with a total loss of KG1a cell viability with 10 µM of the chemotherapeutic drug. In regards to HL60 cells, high cell viability was maintained even after the addition of 0.001 and 0.1 µM of Dox, compared with the untreated HL60 cells. In contrast to the KG1a cells, a total loss of HL60 cell viability was observed with the addition of lower Dox concentrations. The KG1a cells, characterized as CD34$^+$/CD38$^-$ using flow cytometer, had an expected chemotherapeutic resistance potential compared to the HL-60 cell line characterized as CD45$^+$/CD32$^+$/CD52$^-$/CD34$^-$ based on the transcriptomic profiling, consistent with previous studies [28, 33, 34]. The resistance of the KG1a cells to the chemotherapeutic drug may be caused by many mechanisms including efflux, alteration of the drug target and cell death inhibition, such as resistance to apoptosis, the programmed cell death, or any other process involving the drug resistance-related signal pathway [17]. We hypothesized that the blockade of some survival pathways, including the NF-κB pathway, the p38 MAPK survival pathway, and reactive oxygen species (ROS) generation may enhance the sensitivity of the KG1a cells to the chemotherapeutic drugs and overcome multidrug resistance.

Numerous signaling pathways induce multidrug resistance by promoting cell proliferation and inhibiting apoptosis [44–46]. We hypothesized that the blockade of some survival pathways such as NF-kB and p38 MAPK pathways and the production of ROS might increase leukemic cell sensitivity to chemotherapy. To fully understand cell death inhibition resistance, we investigated the impact of pharmacological inhibitors on the viability of the HL60 and KG1a cells after the blockade of ROS generation, NF-κB and p38 MAPK pathways. In general, cancer cells generate a high level of ROS due to the intense cell metabolism contributing to the development and progression of cancer. Although the excessive production of mitochondrial ROS is a critical mediator of genotoxic-induced apoptosis through the activation of the mitochondrial outer membrane permeabilization [47], the levels of cellular ROS generated and the activity of scavenging/anti-oxidant enzymes are higher in cancer than normal cells. A modulation of the cellular ROS generated was suggested to sensitize multidrug resistant cancer cells to certain chemotherapeutic drugs [48]. In addition, the ROS scavenger NAC, the precursor of reduced glutathione and a modulator of the intracellular redox state, induces apoptosis via the intrinsic mitochondrial pathway in cardio-myoblastic H9c2 cells [49]. In the present study, the addition of NAC did not affect the apoptotic status in the AML cell lines KG1a and HL60 or in the presence of the chemotherapeutic drug (S1 Fig). NF-κB is an essential regulator of multiple cellular processes involved in leukemogenesis, including proliferation, differentiation, and apoptosis [50]. Approximately 40% of the AML patients exhibit unusual NF-κB signaling activity that enables myeloid cells to escape apoptosis and stimulate proliferation [51]. A possible therapeutic strategy to cure AML is to inhibit the NF-κB signaling pathway that was targeted to overcome multidrug resistance [52]. We used flow cytometry to determine the apoptotic status of the KG1a and HL-60 cells treated with NF-κB inhibitor (IKKVII). After 72 h incubation, most of the cell population from both cell lines were in late apoptosis and regardless the addition of the chemotherapeutic drug (S1 Fig), compared to the healthy non-treated cells. Similarly, on the blockade of p38 MAPK survival pathway in the AML cell lines with the p38 MAPK inhibitor SB202190, followed by the addition of the chemotherapeutic drug, an increase in the pro-apoptotic cells was observed. Previous studies indicated that the MAPK pathway is involved in developing the chemotherapeutic resistance of cancer cells [53]. We investigated the effect of the p38 MAPK inhibitor SB202190 in drug resistant- and drug-sensitive cell lines, the KG1a and HL60. The addition of chemotherapy drugs to the cells pretreated with SB202190 significantly decreased the phosphorylation level of the p38 MAPK, compared to the p38 MAPK phosphorylation level detected in the non-treated cells, in SB202190-treated cells and in FUrd-treated cells. The cell pretreatment with p38 MAPK inhibitor enhanced the cell sensitivity to FUrd in the KG1a and HL60 via the inhibition of the p38 MAPK pathway.

Apoptosis-related assays were performed for the assessment of effector caspase-3/-7 activity and of mitochondrial outer membrane permeabilization activity, two prominent hallmarks of apoptosis. The cytotoxic effect of the chemotherapeutic drug FUrd was confirmed by the increased caspase- and mitochondria-based apoptotic activities while p38 MAPK blockade did not show a significant effect. However, the addition of FUrd after the p38 MAPK cell pretreatment resulted in an increase of the caspase-3/-7 and mitochondrial outer membrane permeabilization activities, higher than with just FUrd alone. A possible mechanism by which AML patients develop drug resistance is via the constitutive activation of some of the essential cellular survival pathways. These findings confirm the importance of the NF-κB and the p38 MAPK pathways as attractive targets for therapeutic drug development against cancer growth and progression [54, 55]. Of the factors that may modulate key signal transduction pathways, small non-coding RNAs named as microRNAs (miRNA) regulate the expression of specific genes involved in various cellular processes including cancer formation, development, and also in multidrug resistance [56, 57]. The expression profiling of miRNA has been reported to predict the drug resistance of cancer cells [41], hence an establishment of miRNA expression profiling as a validation of the improvement of KG1a cell chemosensitivity after p38 MAPK survival pathway blockade is of interest.

Recently, using high-throughput technology, many studies reported that miRNAs play a role in the chemotherapy resistance in cancer patients via targeting drug resistance-related genes, or genes that control the cell cycle, cell proliferation, and apoptosis [58]. This present study compared the expression levels of miRNAs detected in FUrd-treated chemoresistant KG1a cells with the levels detected in the FUrd-treated chemosensitive KG1a cells (after p38 MAPK blockade). Based on our microarray analysis, the highest upregulation of miRNA expression level monitored in the chemosensitive treated group compared to the chemoresistant treated group corresponded to the miR-328-3p. Associated with a poor prognosis in AML patients, a low circulating miR-328 expression level was found in AML patients' blood samples, which increased in patients after daunorubicin combined with cytarabine treatment and in complete remission, suggesting miR-328 as a potential non-invasive biomarker for cancer diagnosis [59]. Reported to be downregulated in human castration-resistant prostate cancer tissues, the forced miR-328 overexpression in mimic miR-328 transfected-prostate cancer cell line resulted in the enhancement of cell sensitivity to the chemotherapeutic drug docetaxel and in the induction of apoptosis associated with the increase of the cleavage of caspase-3 and caspase-9 (mitochondrial-dependent apoptotic pathway) [42]. Using human osteosarcoma cell lines, the phytochemical resveratrol with antimetastatic properties upregulated miR-328 expression through the p38 MAPK pathway activation [60]. In addition, the miR-328 decreased chemoresistance by targeting several genes involved in the ABC transporter, such as ABCG2 also known as breast cancer resistance protein BCRP, ABCD3 [61, 62]. Suggested as a novel biomarker for the diagnosis and as a promising strategy for therapeutic manipulation, miR-328-3p combined with miR-519d suppressed breast cancer progression by targeting Ki67 [63]. Reported as a potential non-invasive poor prognosis biomarker for AML patients, deeper investigation of the main functions of the miR-328-3p in the AML stem cell chemoresistance capacity, which combined with another miRNA, might downregulate common target genes, which code survival signaling proteins, could eventually be a promising strategy to overcome AML relapse.

Concerning miR-210, a known hypoxia-regulated miR, also referred to as hypoxamiR, plays a regulator role in several cellular functions dependent and independent of hypoxia, a frequent feature of the tumor microenvironment deprived in oxygen [64]. In hypoxic vascular smooth muscle cells, the overexpression of miR-210 resulted in the inhibition of pro-apoptotic Bax, Bad and caspase-3 cleavage, and the downregulation of miR-210 expression promoted

hypoxia-induced apoptosis [64]. Several studies revealed that a higher expression of miR-210 was associated with tumor development and progression, including breast cancer, prostate cancer and AML [43, 65, 66]. Using a cisplatin (DDP)-resistant breast cancer cell line MCF-7/ DDP, miR-210 downregulation reduced the chemoresistance potential and increased apoptosis by targeting the JAK/STAT signaling pathway [67]. These findings align with our data showing the increased AML stem cell death and apoptosis induction associated with the lowest expression level of miR-210 detected in combined (SB202190+FUrd) treated-KG1a cells. However, in bone marrow and in minimal residual disease (MRD) samples, the combined detection of low miR-210 expression and of low expression of caspase 8-associated protein 2 (CASP8AP2), a pro-apoptotic protein suggested a promising prognostic indicator for hematological relapse, was associated with a poor outcome in pediatric acute lymphoblastic leukemia [68]. Despite these conflicting results, monitoring the miR-210 expression level in blood, bone marrow-derived mesenchymal stem cells and in MRD samples of relapsed AML patients in the search for potential apoptosis-related target genes would be of interest for the validation of miR-210 expression level as a potential prognostic biomarker and the development of a therapeutic strategy to reduce the AML relapse rate.

## Conclusions

Targeting the p38 MAPK survival pathway sensitized the chemoresistant AML cell line KG1a in response to the cytotoxic effect of the chemotherapeutic drug, and was revealed by a decrease of the cell proliferation, the induction of apoptosis through the activation of the extrinsic and intrinsic apoptotic pathways, and the dysregulation of certain miRNA reported to be involved in multidrug resistance, cell proliferation and survival. Our findings pave the way for the development of a new therapeutic strategy to treat relapsed AML patients via targeting the p38 MAPK signaling through miRNAs likely to target gene expression and decrease the multidrug resistance potential. The identification of a new therapeutic target, such as miRNA, resulting in the inhibition of the survival pathway target genes such as p38 MAPK and NF-κB could minimize the relapse rate and elevate the effectiveness of conventional chemotherapeutic drugs.

## Supporting information

**S1 Fig. Apoptotic status of KG1a and HL60 cells exposed to 5-Fluorouridine (FUrd) after cell pretreatment with NAC (ROS production inhibitor) and IKKVII (NF-kB pathway inhibitor).** The HL60 (A) and KG1a (B) cells were pre-treated with 5 mM NAC or 20 μM IKK-VII for 2 h incubation then followed by the cell treatment with 10 μM of FUrd. After 72 h incubation, viable, (early and late) apoptotic, and necrotic status of the cells were determined using Apoptosis determination kit. Representative cell scatter plots indicating the percentage of cells determined at each status. Bar graphs showing the results presented as mean ± SD, based on three independent experiments. $^*p < 0.05$, $^{**}p < 0.01$, and $^{****}p < 0.0001$ *vs.* control. (TIF)

**S1 Table. Classification of the 100 genes most differentially expressed by comparing the KG1a (resistant) against the HL60 (sensitive) cell lines.** (XLSX)

**S2 Table. CD marker differentially expressed by comparing the AML stem cell line KG1a against the promyelocytic leukemia cell line HL60.** (XLSX)

**S3 Table. ATP-binding cassette (ABC) differentially expressed by comparing the KG1a against the HL60 cell lines.**
(XLSX)

**S4 Table. Data analysis of the profiling of the top differentially dysregulated drug resistance-associated miRNA in AML stem cell line KG1a exposed to SB202190 p38 MAPK inhibitor compared to untreated KG1a cells.**
(XLSX)

**S5 Table. Data analysis of the profiling of the top differentially dysregulated drug resistance-associated miRNA in AML stem cell line KG1a exposed to SB202190 p38 MAPK inhibitor and to 1 µM FUrd.**
(XLSX)

**S6 Table. Data analysis of the profiling of the top differentially dysregulated drug resistance-associated miRNA in AML stem cell line KG1a exposed to SB202190 p38 MAPK inhibitor and to 10 µM FUrd.**
(XLSX)

## Acknowledgments

We are grateful to Dr Rizwan Ali from KAIMRC Medical Research Core Facility and Platforms for the use of the confocal scanner lasing microscope and for providing us the confocal images.

## Author Contributions

**Conceptualization:** Sabine Matou-Nasri.

**Data curation:** Maria Najdi, Nouran Abu AlSaud, Yazeid Alhaidan, Hamad Al-Eidi, Ghada Alatar, Deemah AlWadaani, Thadeo Trivilegio, Arwa AlSubait, Abeer AlTuwaijri, Manal Abudawood, Bader Almuzzaini.

**Formal analysis:** Maria Najdi, Nouran Abu AlSaud, Yazeid Alhaidan, Ghada Alatar, Deemah AlWadaani, Thadeo Trivilegio, Arwa AlSubait, Abeer AlTuwaijri, Bader Almuzzaini.

**Funding acquisition:** Sabine Matou-Nasri.

**Investigation:** Sabine Matou-Nasri, Bader Almuzzaini.

**Methodology:** Sabine Matou-Nasri, Maria Najdi, Nouran Abu AlSaud, Yazeid Alhaidan, Hamad Al-Eidi, Ghada Alatar, Deemah AlWadaani, Thadeo Trivilegio, Arwa AlSubait, Abeer AlTuwaijri, Manal Abudawood, Bader Almuzzaini.

**Project administration:** Deemah AlWadaani, Manal Abudawood, Bader Almuzzaini.

**Resources:** Nouran Abu AlSaud, Ghada Alatar, Bader Almuzzaini.

**Software:** Maria Najdi, Nouran Abu AlSaud, Yazeid Alhaidan, Hamad Al-Eidi, Ghada Alatar, Deemah AlWadaani, Thadeo Trivilegio, Arwa AlSubait, Abeer AlTuwaijri, Manal Abudawood, Bader Almuzzaini.

**Supervision:** Sabine Matou-Nasri, Bader Almuzzaini.

**Validation:** Sabine Matou-Nasri, Maria Najdi, Nouran Abu AlSaud, Yazeid Alhaidan, Hamad Al-Eidi, Ghada Alatar, Deemah AlWadaani, Thadeo Trivilegio, Arwa AlSubait, Abeer AlTuwaijri, Manal Abudawood, Bader Almuzzaini.

**Visualization:** Maria Najdi, Nouran Abu AlSaud, Yazeid Alhaidan, Hamad Al-Eidi, Ghada Alatar, Deemah AlWadaani, Thadeo Trivilegio, Arwa AlSubait, Abeer AlTuwaijri, Manal Abudawood, Bader Almuzzaini.

**Writing – original draft:** Sabine Matou-Nasri, Maria Najdi.

**Writing – review & editing:** Sabine Matou-Nasri, Maria Najdi, Nouran Abu AlSaud, Yazeid Alhaidan, Hamad Al-Eidi, Ghada Alatar, Deemah AlWadaani, Thadeo Trivilegio, Arwa AlSubait, Abeer AlTuwaijri, Manal Abudawood, Bader Almuzzaini.

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
