## [Decision Letter · Decision Letter 0]

21 Mar 2022

PONE-D-22-03405Blockade of p38 MAPK overcomes AML stem cell line KG1a resistance to 5-Fluorouridine and the impact on miRNA profilingPLOS ONE

Dear Dr. Matou-Nasri,

Thank you for submitting your manuscript to PLOS ONE. After careful consideration, we feel that it has merit but does not fully meet PLOS ONE’s publication criteria as it currently stands. Therefore, we invite you to submit a revised version of the manuscript that addresses the points raised during the review process. Specifically, reviewer 1 has identified some major and minor issues that require further attention before the manuscript can be published. Please carefully address all these issues in the revised manuscript.

We look forward to receiving your revised manuscript.

Kind regards,

Jinsong Zhang

Academic Editor

PLOS ONE

Journal Requirements:

(NO authors have competing interests)

5. Please include your tables as part of your main manuscript and remove the individual files. Please note that supplementary tables (should remain/ be uploaded) as separate "supporting information" files".

Reviewers' comments:

Reviewer's Responses to Questions

**Comments to the Author**

1. Is the manuscript technically sound, and do the data support the conclusions?

Reviewer #1: Yes

Reviewer #2: Yes

2. Has the statistical analysis been performed appropriately and rigorously? 

Reviewer #1: Yes

Reviewer #2: Yes

3. Have the authors made all data underlying the findings in their manuscript fully available?

Reviewer #1: Yes

Reviewer #2: Yes

4. Is the manuscript presented in an intelligible fashion and written in standard English?

Reviewer #1: Yes

Reviewer #2: Yes

5. Review Comments to the Author

Reviewer #1: This quite interesting manuscript by Sabine Matou-Nasri et al. demonstrates that P38 MAPK pathway blockade enhanced KG1α cell sensitivity to 5-Fluorouridine, which was associated with the up-regulation of miR-328-5p and down-regulation of miR-210-5p.

Comment 1: In this research, mostly all the KG1α cells were positive in stem cell marker CD34 and negative in lymphocyte differentiation marker CD38. For the results, the authors should provide statistical diagrams, at least as supplemental materials. However, other studies had shown that a significant number of KG1α cells were not CD34+/CD38-. The proportion of CD34+/CD38- cells in these studies ranged from 17.6%±1.2% to 54.167%±6.57%. Maybe the enrichment of CD34+/CD38- LSCs using magnetic beads could help to find the main gene expression profiling characterizing the chemoresistance potential of AML stem cell.

Comment 2: HL60 is known as chemosensitive AML-M3 cell line. However, 5FUrd is rarely used in the treatment of acute promyelocytic leukemia. Although it is common to select HL60 as the control cell of KG1α, it is not appropriate to use HL60 in the study of 5FUrd.

Comment 3: In the abstract and discussion, the authors wrongly described‘miR-328-3p’ as ‘miR-328-5p’ .

Reviewer #2: The manuscript titled "Blockade of p38MAPK overcomes AML stem cell line KG!a..." by Matou-Nasri S., et al. concerns the drug resistance phenomenon, which often occurs during the treatment of AML patients. The researchers' attention is drawn to AML stem cells due to their high potential for chemoresistance.

The authors of this study showed that targeting p38 MAPK pathway can sensitize drug-resistant AML cells in the KG1a cell line. This drug sensitivity effect was also dependent on miRNA downregulation. The work is done very carefully and shows that by blocking these two signaling pathways, it induces apoptosis in KG1a cells in both extrinsic and intrinsic manner as well as inhibition of the cell proliferation.

Data shown in the manuscript are very solid and convincing, clearly proves that the thesis and conclusions contained in this paper are true and relevant.

In my opinion it should be accepted for publication in PLOS ONE as it is now.

6. PLOS authors have the option to publish the peer review history of their article (what does this mean?). If published, this will include your full peer review and any attached files.

Reviewer #1: No

Reviewer #2: No

---

## [Author Response · Author response to Decision Letter 0]

30 Mar 2022

Dear Dr Jinsong Zhang,

Thank you for considering this manuscript for review. We are grateful to all reviewers for their detailed feedback and believe that this has improved the quality of our revised manuscript. Wherever possible, our paper has been modified in light of the comments made by the reviewers. Please find below our detailed response (in red) to each point raised by the reviewers.

Reviewer #1: This quite interesting manuscript by Sabine Matou-Nasri et al. demonstrates that P38 MAPK pathway blockade enhanced KG1α cell sensitivity to 5-Fluorouridine, which was associated with the up-regulation of miR-328-5p and down-regulation of miR-210-5p.

Comment 1: In this research, mostly all the KG1α cells were positive in stem cell marker CD34 and negative in lymphocyte differentiation marker CD38. For the results, the authors should provide statistical diagrams, at least as supplemental materials. However, other studies had shown that a significant number of KG1α cells were not CD34+/CD38-. The proportion of CD34+/CD38- cells in these studies ranged from 17.6%±1.2% to 54.167%±6.57%. Maybe the enrichment of CD34+/CD38- LSCs using magnetic beads could help to find the main gene expression profiling characterizing the chemoresistance potential of AML stem cell.

Thank you for your comment. As suggested, a bar graph summarizing the average of the percentage of KG1a cells positive in CD34 and negative in CD38 based on three independent analyses has been added to the Figure 1A. The Figure legends has been modified as well. We found the study that you referred to, which reported that “KG1a cells contained (54.167+6.57)% of CD34+CD38-“ (She et al. Cancer Letters 318(2012)173-179). However, the authors did not mention the source of these KG1a cells. As mentioned in our study, the used KG1a cells are commercialized cells (#CCL-246.1) from ATCC. In agreement with our FACS analysis confirming the ATCC-CCL-246.1 homogeneous KG1a cell population presenting the chemoresistant phenotype CD34+/CD38-, recent studies reported the full KG1a cell population were positive in CD34 (Arruda et al. Haematologica 2022 PMID:35142149. DOI: 10.3324/haematol.2021.279486) and negative in CD38 (Gurney et al. Haematologica 2022. 107(2):437-445). 

Comment 2: HL60 is known as chemosensitive AML-M3 cell line. However, 5FUrd is rarely used in the treatment of acute promyelocytic leukemia. Although it is common to select HL60 as the control cell of KG1α, it is not appropriate to use HL60 in the study of 5FUrd.

That is correct that 5FUrd is rarely used in the treatment of acute promyelocytic leukemia patients. However, several studies reported new insights into the molecular mechanisms underlying the cancer cell response to chemotherapeutic drugs after examining the anticancer effect of 5-Fluorouridine on HL60 cells, especially at the transcriptional level (Grant et al. Cancer Research 44 (1984) 5505-5510; Zlatopolskiy et al. J Nucl Med 50 (2009) 1895-1903; Maeharma et al. J Biol Chem. 289 (2014) 20802-20812). Hence, our present study still provides important insight related to the transcriptional machinery of HL60 cells (i.e. miRNA profiling) in response to 5-FUrd. 

Comment 3: In the abstract and discussion, the authors wrongly described‘miR-328-3p’ as ‘miR-328-5p’ .

Thank you for this observation. The corrections have been made.

Reviewer #2: The manuscript titled "Blockade of p38MAPK overcomes AML stem cell line KG!a..." by Matou-Nasri S., et al. concerns the drug resistance phenomenon, which often occurs during the treatment of AML patients. The researchers' attention is drawn to AML stem cells due to their high potential for chemoresistance.

The authors of this study showed that targeting p38 MAPK pathway can sensitize drug-resistant AML cells in the KG1a cell line. This drug sensitivity effect was also dependent on miRNA downregulation. The work is done very carefully and shows that by blocking these two signaling pathways, it induces apoptosis in KG1a cells in both extrinsic and intrinsic manner as well as inhibition of the cell proliferation.

Data shown in the manuscript are very solid and convincing, clearly proves that the thesis and conclusions contained in this paper are true and relevant.

In my opinion it should be accepted for publication in PLOS ONE as it is now.

Thank you for this positive comment. Really appreciated, indeed.

Of note, in accordance with PLOS requirements, “data not shown” has been deleted and a Supplementary file of relevant data has been included. 

We hope these changes are satisfactory as we are keen to publish in PLoS One Journal. Please feel free to contact us should you require further clarification. 

We are looking forward to hearing from you

With my best regards

Sabine Matou-Nasri, PhD

Senior research scientist

Cell and Gene Therapy Group

Medical Genomics Research Department 

King Abdullah International Medical Research Center (KAIMRC)

---

## [Decision Letter · Decision Letter 1]

18 Apr 2022

Blockade of p38 MAPK overcomes AML stem cell line KG1a resistance to 5-Fluorouridine and the impact on miRNA profiling

PONE-D-22-03405R1

Dear Dr. Matou-Nasri,

We’re pleased to inform you that your manuscript has been judged scientifically suitable for publication and will be formally accepted for publication once it meets all outstanding technical requirements.

Kind regards,

Jinsong Zhang

Academic Editor

PLOS ONE

Additional Editor Comments (optional):

Reviewers' comments:

Reviewer's Responses to Questions

**Comments to the Author**

1. If the authors have adequately addressed your comments raised in a previous round of review and you feel that this manuscript is now acceptable for publication, you may indicate that here to bypass the “Comments to the Author” section, enter your conflict of interest statement in the “Confidential to Editor” section, and submit your "Accept" recommendation.

Reviewer #1: All comments have been addressed

2. Is the manuscript technically sound, and do the data support the conclusions?

Reviewer #1: Yes

3. Has the statistical analysis been performed appropriately and rigorously? 

Reviewer #1: Yes

4. Have the authors made all data underlying the findings in their manuscript fully available?

Reviewer #1: Yes

5. Is the manuscript presented in an intelligible fashion and written in standard English?

Reviewer #1: Yes

6. Review Comments to the Author

Reviewer #1: This quite interesting manuscript by Matou-Nasri S. et al. demonstrates that P38 MAPK pathway blockade enhanced KG1α cell sensitivity to 5-Fluorouridine, which was associated with the up-regulation of miR-328-5p and down-regulation of miR-210-5p. The author have critically answered all the reviewers’ questions point to point in the rebuttal letter. After revision, the manuscript meets the publication standard of PLOS ONE.

7. PLOS authors have the option to publish the peer review history of their article (what does this mean?). If published, this will include your full peer review and any attached files.

Reviewer #1: No

---

## [Editor Report · Acceptance letter]

21 Apr 2022

PONE-D-22-03405R1 

Blockade of p38 MAPK overcomes AML stem cell line KG1a resistance to 5-Fluorouridine and the impact on miRNA profiling 

Dear Dr. Matou-Nasri:

I'm pleased to inform you that your manuscript has been deemed suitable for publication in PLOS ONE. Congratulations! Your manuscript is now with our production department. 

Kind regards, 

on behalf of

Dr. Jinsong Zhang 

Academic Editor

PLOS ONE